# Structural Adaptation of the Excitation–Contraction Coupling Apparatus in Calsequestrin1-Null Mice during Postnatal Development

**DOI:** 10.3390/biology12081064

**Published:** 2023-07-29

**Authors:** Stefania Murzilli, Matteo Serano, Laura Pietrangelo, Feliciano Protasi, Cecilia Paolini

**Affiliations:** 1Department of Neuroscience, Imaging and Clinical Sciences (DNISC), Center for Advanced Studies and Technology (CAST), University G. d’Annunzio of Chieti-Pescara, 66100 Chieti, Italy; stefania.murzilli@gmail.com; 2Department of Medicine and Aging Sciences (DMSI), Center for Advanced Studies and Technology (CAST), University G. d’Annunzio of Chieti-Pescara, 66100 Chieti, Italy; matteo.serano@unich.it (M.S.); laura.pietrangelo@unich.it (L.P.); feliciano.protasi@unich.it (F.P.)

**Keywords:** skeletal muscle, EC coupling, calsequestrin, development

## Abstract

**Simple Summary:**

The efficiency of the contractile machinery in muscle fibers is affected by the adequate development of skeletal muscle. CASQ1 knockout impairs the correct assembly of the excitation–contraction coupling apparatus and functional properties. In this study, to obtain further clues on CASQ1 functions, we characterized the ultrastructural alterations and examined the maturation of calcium release units, as well as the involvement of different junctional proteins in juxtaposition of the membrane system in skeletal muscle fibers during development from birth to the adult age. In the calcium release units, junctional proteins directly linked to CASQ1 exhibit a modified expression pattern during postnatal development, resulting in a delayed maturation process. In summary, our findings demonstrate that CASQ1 is essential for the full maturation of calcium release units during postnatal development. The disruption caused by its absence affects the correct assembly and coordination of junctional proteins, resulting in impaired Transverse-tubule biogenesis and altered muscle morphology. This study enhances our understanding of the intricate mechanisms underlying calcium release units formation and highlights the critical role of CASQ1 in this process.

**Abstract:**

The precise arrangement and peculiar interaction of transverse tubule (T-tubule) and sarcoplasmic reticulum (SR) membranes efficiently guarantee adequate contractile properties of skeletal muscle fibers. Fast muscle fibers from mice lacking calsequestrin 1 (CASQ1) are characterized by the profound ultrastructural remodeling of T-tubule/SR junctions. This study investigates the role of CASQ1, an essential component of calcium release units (CRUs), in the postnatal development of muscle fibers. By using CASQ1-knockout mice, we examined the maturation of CRUs and the involvement of different junctional proteins in the juxtaposition of the membrane system. Our morphological investigation of both wild-type (WT) and CASQ1-null extensor digitorum longus (EDL) fibers, from 1 week to 4 months of age, yielded noteworthy findings. Firstly, we observed that the absence of CASQ1 hindered the full maturation of CRUs, despite the correct localization of key junctional components (ryanodine receptor, dihydropyridine receptor, and triadin) to the junctional SR in adult animals. Furthermore, analysis of protein expression profiles related to T-tubule biogenesis and organization (junctophilin 1, amphiphysin 2, caveolin 3, and mitsugumin 29) demonstrated delayed progression in their expression during postnatal development in the absence of CASQ1, suggesting the impaired maturation of CRUs. The absence of CASQ1 directly impacts the proper assembly of CRUs during development and influences the expression and coordination of other proteins involved in T-tubule biogenesis and organization.

## 1. Introduction

Excitation–contraction (EC) coupling, the process that triggers the release of Ca^2+^ from the sarcoplasmic reticulum (SR) during muscle contraction [1,2], occurs at highly specialized intracellular junctions, namely, Ca^2+^-release units (CRUs). CRUs are formed via the association of plasma membrane invaginations, called transverse-tubules (T-tubules), with the SR terminal cisternae. In adult mouse skeletal muscle fibers, CRUs take the form of triads, membrane complexes that are each composed of one T-tubule and two SR terminal cisternae [3]. During muscle postnatal differentiation, the EC coupling machinery undergoes changes both in the morphological architecture [4] and at a molecular level [5,6,7]. At birth, the T-tubules are still predominantly longitudinally oriented, there are few CRUs and they are not yet associated with the cross striation created by the lateral alignment of sarcomeres. However, already around the first week of age, a conspicuous number of triads is positioned at the A–I junctions of sarcomeres, as in mature skeletal muscle. Around 10 days of age, most of the T-tubule network and the CRUs are transversally oriented, and maturation of the two membrane systems is complete at about 3 weeks after birth. Some intracellular junctions, though, still present features of immature muscle, i.e., they have a longitudinal orientation and/or are multi-layered (e.g., pentads).

Several junctional proteins have been shown to participate in the postnatal maturation of SR and T-tubule membrane systems: caveolin 3 (CAV3), associated with the formation of sarcolemma invaginations and developing T-tubules [8]; amphiphysin 2/Bridging Integrator-1 (BIN1), which promotes membrane tubulation [9]; mitsugumin 29 (MG29), which promotes T-tubule elongation [10]; and finally junctophilins (JPs), which act as molecular bridges between T-tubule and SR membranes [11,12,13]. It has been found that knockout of junctophilin 1 (JP1) in skeletal muscle fibers is responsible for the reduced number of triads [14,15], but that, on the other hand, the overexpression of JP2 is responsible for a significant increase in multiple SR–T-tubule contacts, although other EC coupling-protein levels are not changed [16].

Once the SR and T-tubule membranes are assembled, control of SR Ca^2+^ release during EC coupling is coordinated by other proteins; the dihydropyridine receptors (DHPRs) placed in the T-tubule membrane function as voltage sensors during membrane depolarization [1,2], which mechanically activate the ryanodine receptors (RyRs), i.e., Ca^2+^-release channels located in the opposing membrane of the SR. Mammalian adult fast skeletal muscle fibers contain only isoform 1 of RyR (RyR1); on the other hand, during the late embryonic stage and in slow adult skeletal muscle fibers, the predominant RyR1 isoform is co-expressed with low levels of RyR3. For reviews, see Sorrentino and Volpe (1993) [17], Sorrentino (1995) [18], Block et al. (1996) [19], Sutko and Airey (1997) [20], and Franzini-Armstrong and Protasi (1997) [21]. The differential expression of RyR isoforms may be important for generating specific patterns of Ca^2+^ signaling [22]. Several other proteins located in CRUs are involved in this mechanism: calsequestrin (CASQ), triadin, and junctin [23,24]. Abnormalities in the expression levels and/or mutations in proteins involved in EC coupling results in the altered control of intracellular Ca^2+^, which may lead to muscle dysfunction and disease [25,26]. CASQ is a protein that binds large amounts of Ca^2+^ within the SR and concentrates it at the junctional face of the terminal cisternae near the sites of Ca^2+^ release, i.e., RyRs [27,28]. There is a general consensus that CASQ is not only important for the ability of the SR to store Ca^2+^, but also for modulating Ca^2+^ release from RyRs [29,30]. Structural studies suggest that CASQ is indeed in the correct location to control the activity of the Ca^2+^-release channels [31,32]. CASQ1 and CASQ2 are two isoforms of CASQ that are expressed in skeletal muscle. Although they share similarities, there are several key differences between these isoforms. CASQ1 is primarily found in skeletal muscle, whereas CASQ2 is predominantly expressed in cardiac muscle. However, CASQ2 can also be present in low levels in skeletal muscle, particularly during fetal and neonatal stages. CASQ2 is the most abundant isoform during fetal and neonatal stages [7]. However, as the muscle matures into adulthood, CASQ2 becomes less prevalent, accounting for only about 25% of the total CASQ present. On the other hand, CASQ1 steadily and rapidly increases in amount during postnatal development and becomes the predominant isoform in adult skeletal muscle [33]. In fast-twitch fibers, CASQ2 gradually disappears during postnatal development and at two to four weeks postnatally is almost totally replaced by the skeletal isoform CASQ1 that steadily and quickly increases in amount [7]. However, it is relevant to consider that a more sensitive technique, such as the proteomic technique, is able to detect both CASQ isoforms in mature skeletal muscle [34]. Triadin and junctin have been proposed to represent the structural/functional link between CASQ and RyR1 and may contribute to modulate Ca^2+^-channel activity [23,24].

In our laboratory, we characterized a knockout model lacking CASQ1: CASQ1-null mice [35,36]. The lack of CASQ1 causes striking remodeling of CRUs (more evident in fast-twitch than in slow-twitch muscles) with a consequent partial impairment of the EC coupling mechanism. In adult wild type (WT) fibers, the T-tubules are usually associated with two SR terminal cisternae to form triads (i.e., CRUs), whereas in CASQ1-null fibers, CRUs are often formed by multiple elements due to the striking proliferation of both SR and T-tubules. The junctional SR profile of multi-layered CRUs is decorated with several rows of RyRs/feet instead of only two rows observed in WT triads. Finally, unlike in WT fibers, where the T-tubule network exhibits a transverse orientation, CRUs in CASQ1-null fibers are frequently longitudinal and oblique. This partial longitudinal disposition of the EC coupling system suggests that the maturation never reaches a complete maturation state [35], and it is even possible that the non-proper disposition of CRUs is due to the lack of transversal space.

Here we performed an accurate examination of postnatal development of the EC coupling apparatus in CASQ1-null and WT fast-twitch fibers: (1) we analyzed the expression of junctional proteins known to be responsible for CRU formation (JP1, BIN1, CAV3, and MG29); (2) we followed the T-tubular network maturation in CASQ1-null fibers at different ages (from 1 week to 4 months of age); and finally (3) we performed a detailed quantitative analysis of the frequency during the development of different type of CRU (longitudinal, transverse, oblique, and multiple) junctions.

## 2. Materials and Methods

### 2.1. CASQ1-Null Mice

CASQ1-null mice were generated as previously described [35]. Mice were housed in micro-isolator cages, temperature 20 °C, 12 h light/dark cycle, with free access to water and food. The initial characterizations of functional and structural phenotypes of CASQ1-null muscles were published in Paolini et al. (2007) [36]. All experiments on animals were conducted according to the Directive of the European Union 2010/63/UE and approved by the Committee on the Ethics of Animal Experiments of the University of Chieti (50/2012/CEISA/COM).

### 2.2. Preparation of Homogenate Total Membranes, Electrophoresis, and Western Blot Analysis

Preparation of total homogenates from EDL muscles of WT and CASQ1-null at different ages and Western blot analysis were performed as previously described [35]. Protein concentration was determined spectrophotometrically using a Pierce BCA Protein Assay Kit (ThermoFisher scientific, Waltham, MA, USA). Of the total protein, 20–40 µg was resolved in 8% or 10% SDS–polyacrylamide gel, transferred to nitrocellulose membrane, and blocked with 5% nonfat dry milk (EuroClone, Milan, Italy) in Tris-buffered saline 0.1% and Tween 20 (TBS-T; Sigma-Aldrich, St. Louis, MO, USA) for 1 h. Membranes were probed with the following primary antibodies diluted in 5% nonfat dry milk in TBS-T overnight at 4 °C: rabbit polyclonal against both CASQ1 and CASQ2, diluted 1:1000 (PA1-913, ThermoFisher Scientific, Waltham, MA, USA); rabbit polyclonal anti-BIN1, diluted at 1:1000 (Ab153912, Abcam, Waltham, MA USA); JP1, diluted 1:500 (405100, ThermoFisher Scientific, Waltham, MA, USA); mouse monoclonal anti-CAV3, diluted 1:1000 (Sc-5310, Santa Cruz Biotechnology, Inc., Dallas, TX, USA); goat polyclonal anti-MG29, diluted 1:2000 (Sc-23441, Santa Cruz Biotechnology, Inc., Dallas, TX, USA); mouse monoclonal anti-RyR1, diluted 1:30 (34C, Developmental Studies Hybridoma Bank, University of Iowa, IA, USA); and mouse monoclonal anti-glyceraldehyde-3-phosphate dehydrogenase (GAPDH) antibody, diluted 1:5000 (802519, OriGene Technologies, Rockville, MD, USA), used as a loading control. Membranes were washed 3 times in TBS-T and incubated with goat, mouse, and rabbit secondary antibodies (horseradish peroxidase conjugated, Millipore, Merck Millipore, Burlington, MA, USA), diluted 1:10,000 in 5% nonfat dry milk in TBS-T for 1 h at room temperature. Proteins were detected by ECL liquid (PerkinElmer, Waltham, MA, USA), and quantification was conducted using an Alliance imaging system and software (Uvitec Ltd., Cambridge, UK). Original western blot images are visible in Appendix A.

### 2.3. Real-Time PCR

Total RNA extracted from WT and CASQ1-null pools of 10 EDL muscles from n = 5 mice, respectively, at week 1 and at 1 and 4 months of age, was isolated using Trizol^®^ Reagent (ThermoFisher Scientific, Waltham, MA, USA) and treated with deoxyribonuclease I (Ambion, Austin, TX, USA) according to the manufacturer’s protocol. The RNA concentration was measured using a NanoDrop 2000c spectrophotometer (ThermoFisher Scientific, Waltham, MA, USA), and the integrity was assessed by examination on agarose gel electrophoresis. Then 4 µg of total RNA was reverse transcribed in a total volume of 100 μL using a high-capacity cDNA reverse transcription kit (Applied Biosystem, Waltham, MA, USA). Primers to detect the mRNA expression levels of each gene were designed using Primer Express software (Applied Biosystems, Waltham, MA, USA) based on GenBank sequence data: mouse glyceraldehyde-3-phosphate dehydrogenase (GAPDH, Mm 99999915-g1), mouse calsequestrin 1 (CASQ1, Mm 00486725-m1), mouse calsequestrin 2 (CASQ2, Mm 00486742-m1), mouse junctophilin 1 (JPH1, Mm 00517485-m1), mouse bridging integrator 1 (BIN1, Mm 00437457-m1), mouse caveolin 3 (CAV3, Mm 01182632-m1), and mouse synaptophysin-like 2 (SYLP2/MG29, Mm 00485005-m1). Quantitative reverse-transcription polymerase chain reaction assays were performed in 96-well optical reaction plates using the ABI Prism 7900 Sequence Detection System (ThermoFisher Scientific, Waltham, MA, USA). Polymerase chain reaction assays were conducted in triplicate wells for each sample. Baseline values of amplification plots were set automatically, and threshold values were kept constant to obtain normalized cycle times and linear regression data. The reaction mixture per well used was as follows: 10 μL Power SYBR Green (Applied Biosystems, Waltham, MA, USA), 2.4 μL of primers at a final concentration of 150 nmol/L, 4.6 μL ribonuclease-free water, and 3 μL complementary DNA. For all experiments, the following polymerase chain reaction conditions were used: denaturation at 95 °C for 10 min, followed by 40 cycles at 95 °C for 15 s and then at 60 °C for 60 s. Quantitative normalization of complementary DNA in each sample was performed using GAPDH as an internal control. Relative gene expression was calculated using the comparative 2^−ΔC^_t_ or 2^−ΔΔC^_t_ methods. The mRNA expression profile from CASQ1-null EDL muscles at week 1 and at 1 and 4 months of age was expressed as fold change relative to the counterpart WT EDL muscles, to which were assigned a value of 1.0.

### 2.4. Preparation of Samples for Electron Microscopy (EM)

EDL muscles were carefully dissected from WT and CASQ1-null mice from three main age groups: 1 week, 1 month, and 4 months of age. Muscles were fixed at room temperature (RT) in 3.5% glutaraldehyde in 0.1M sodium cacodylate buffer, pH 7.2, for 2 h and kept in fixative until further use. (*a*) *Transmission electron microscopy (TEM).* Small bundles of fixed fibers were post-fixed and embedded as in Paolini et al. (2011) [36]. Ultrathin sections were cut (45 nm) and, after staining with 4% uranyl acetate and lead citrate, examined with a Morgagni Series 268D electron microscope (FEI Company, Brno, Czech Republic) equipped with a Megaview III digital camera. (*b*) *Freeze-fracture.* Small pieces of fixed muscle were cryoprotected with 30% (*v*/*v*) glycerol, placed on a gold holder, and frozen in liquid-nitrogen-cooled propane (see Paolini et al., 2004, for more detail [37]). The fractured surfaces were shadowed with platinum at 45° unidirectionally and replicated with carbon in a freeze-fracture apparatus (model BFA 400; Balzers S.p.A., Hudson, NH, USA).

### 2.5. Immunohistochemistry

EDL muscles were dissected from both WT and CASQ1-null mice (4–6 months of age) and fixed at RT in 2% paraformaldehyde in 1X phosphate-buffered saline (PBS). Small bundles of fixed fibers, cleaned of any collagen and fibrous tissue, were washed 3 times in PBS buffer plus 1% bovine serum albumin (BSA; PBS/BSA). To permeabilize and avoid nonspecific detection, the bundles were treated for 1 h with a permeabilization solution (containing PBS 1X/BSA 1%/goat serum 10% supplemented with Triton X-100 0.5%) at RT, followed by an overnight incubation at 4 °C with primary antibodies (see below for antibodies used and dilution). Bundles were then washed 3 times with PBS/BSA buffer and incubated with secondary antibodies for 1 h at RT before being mounted on coverslips with anti-bleach media (SlowFade Gold Antifade Mountant; ThermoFisher Scientific, Waltham, MA, USA). Code, specificity, working dilution, and the sources of primary antibodies used in single staining experiments were as follows: mouse monoclonal anti-RyR 34C, diluted at 1:30 [38] (Developmental Studies Hybridoma Bank, The University of Iowa, USA); mouse monoclonal anti-α_1S_DHPR, diluted at 1:500 (ThermoFisher Scientific, Waltham, MA, USA); rabbit polyclonal against both CASQ1 and CASQ2, diluted at 1:800 [39]; rabbit affinity-purified TRN6 antibody raised against residues 146–160 of mouse triadin [40], diluted at 1:200. Secondary antibodies used were as follows: Cy3-labeled goat anti-mouse IgG (1:200); and Cy5-labeled goat anti-rabbit IgG (1:200) (Jackson ImmunoResearch Laboratories, West Grove, PA, USA). Images were acquired using a Zeiss LSM510 META laser-scanning confocal microscope system (Zeiss, Jena, Germany) equipped with a Zeiss Axiovert 200 inverted microscope and a Plan Neofluar oil-immersion objective (363/1.3 NA).

### 2.6. Measurements

Measurements of the number of different types of junctions in WT and CASQ1-null mice were performed using transmission electron micrographs. Quantitative analyses were performed on two different time points (1 and 4 months of age). The total number of junctions was counted. The different junctions were classified as multiple, longitudinal, oblique, and traditional transverse triads, and then they were marked and counted. Their average number per 100 µm^2^ was calculated. 

### 2.7. Statistical Analysis

According to data, statistics were expressed as mean ± standard deviation (SD) or as mean ± standard error (SE). Student’s unpaired *t* test was used for comparisons between WT and CASQ1-null data, and statistical significance was set at *p* < 0.05 or *p* < 0.001 for low and high level of significance, respectively.

## 3. Results

### 3.1. CASQ Expression during Development in CASQ1-Null Mice

The postnatal expression of the two CASQ isoforms has been previously shown in rabbit skeletal muscles [7]. Using Western blot analysis, we studied the CASQ expression during development in fast-twitch fibers of CASQ1-null mice. In WT fibers, the turning off of the synthesis of CASQ2, which was abundant in fetal and neonatal stages, took place in the first month after birth. Moreover, the accumulation of CASQ1 increased to reach adult levels around two months of age. In adults, the CASQ2 disappeared almost completely. In CASQ1-null EDL fibers, the CASQ1 expression was obviously missing, while the pattern of expression of CASQ2 resembled that of WT animals (Figure 1A). The expression curves for both WT and CASQ1-null fibers are superimposable (Figure 1B). Densitometric data in Figure 1B are given as relative amount of CASQ (CASQ1 or CASQ2) normalized to GAPDH in WT and CASQ1-null fast-twitch fibers.

The mRNA abundance of CASQ1 and CASQ2 genes in EDL muscles isolated from WT and CASQ1-null mice, studied using real-time PCR, confirmed the same pattern of expression of the correspondent active protein following the post-transcriptional regulation mechanisms (Figure 1C,D). Considering that CASQ protein expression did not vary considerably between 2 weeks and 1 month of age, in the subsequent experiments only three time points were considered: 1 week, 1 month, and 4 months of age.

### 3.2. Morphological Features of the Sarcotubular System during the Maturation and Differentiation Process (from 1 Week to 4–6 Months of Age)

To determine differences during development of the transverse (T) tubular network in WT and CASQ1-null fast-twitch fibers, the selective staining of T-tubules in skeletal muscle was performed. In WT fibers 1 week after birth, the T-tubular network showed mainly a longitudinal orientation (arrowheads in Figure 2A), typical of this stage of development.

In fact, it has been demonstrated that in mice, maturation is not complete until about three weeks after birth [4] and that the transverse components of the T-tubule network develop later, with loss of the longitudinal extensions, and invagination of the T-tubules at the junction between the A–I band transition. At 1 month of age, WT EDL fibers were fully developed, and the longitudinal branches of the T-tubular network had almost disappeared compared to the earlier stages of maturation (arrowheads in Figure 2B). Triads were disposed transversally at the A–I junctional regions (small arrows in Figure 2B). Conversely, in CASQ1-null EDL fibers at the same age (1 month), the overall organization of the T-tubular network still appeared to be immature: longitudinal oriented T-tubules (arrowheads) were still frequent despite the age, and multilayered junctions increased rapidly in frequency (arrow in Figure 2C, see also Table 1 for more details).

To confirm the qualitative observations using EM (see above), we performed a detailed quantitative analysis of the frequency of the different type of CRUs during development (Table 1), as represented in Figure 3. CASQ1-null fibers at 1 month of age showed a doubled number of longitudinal junctions and a three-fold increase in the number of multilayered CRUs compared to age-matched WT. Interestingly, the overall number of SR/T-tubule junctions was lower. In 1-month-old WT fibers, most triads already had a transverse orientation, despite there still being a large number of oblique junctions, an unmistakable sign of ongoing maturation. On the other hand, CASQ1-null fibers, even with a similar number of oblique junctions, presented quite a low number of transversal triads compared to WT fibers. At the same time, the number of multiple junctions at 4 months of age increased dramatically, while the number of oblique junctions decreased.

We already demonstrated that one of the compensatory effects of CASQ1 knockout was the striking proliferation of the SR junctional domains to form multiple layers of alternating SR and T-tubule profiles, instead of the classical triad usually found in adult skeletal muscle fibers [35]. In CASQ1-null EDL fibers at 4 months of age, when muscle is considered mature, a delay in the maturation process was still visible. The T-tubular network was not fully reorganized and was split in correspondence with CRUs to form abnormal multilayered junctions (arrows in Figure 2D; see Paolini et al., 2007, for more details [36]). In Paolini et al. (2007), we already showed the appearance, the distribution at the A–I junction, and the orientation of triads in adult CASQ1-null fibers that look more similar to immature WT than to the age-matched samples [35]. Features such as longitudinal junctions containing multiple rows of RyRs/feet, so frequent in young CASQ1-null fibers and still present in a high percentage even in adult CASQ1-null EDL fibers, have been described as immature stages of CRUs/triads (see Flucher et al., 1993 [41] and Protasi et al., 1997 [21]). Furthermore, Takekura and collaborators [42] have shown that early stages of development of the EC coupling apparatus in WT skeletal muscle are accompanied by a progressive penetration of T-tubules deeply into muscle cells and, thus, by a sequential transition from peripheral to internal transverse CRUs. The presence of peripheral couplings, junctions between the SR and surface membranes, was predominant in immature WT fibers (Figure 4A), although in adult CASQ1-null fibers (4 months of age), peripheral couplings were still present (Figure 4B). 

Moreover, patches of SR bearing RyRs/feet arranged in an extended, coherent, tetragonal array that covered the entire junctional SR membrane, resembling the peripheral coupling, were also observed in CASQ1-null fibers at the A–I junction position, where in adult WT muscle, a triad would be located (Figure 4C,D). Freeze-fractures allowed us to visualize the organization of DHPRs; DHPRs in peripheral couplings were organized in tetrad arrays (Figure 4E and large inset), which is the structural base for mechanical/skeletal EC coupling. Particles of DHPRs were located on slightly domed membrane patches (first shown by Takekura and colleagues [42]) and organized in orthogonal arrays. Insets in Figure 4E show details of single tetrads, both complete and incomplete (three-particle) tetrads [43].

### 3.3. Disposition at the Junction of the Sarcomeric Proteins in WT and CASQ1-Null Mice

The morphological alterations in the CASQ1-null fibers of adult mice (between 4 and 6 months of age) were confirmed with immunolabeling assays of sarcomeric proteins. The labeling with specific antibodies performed in CASQ1-null EDL fibers to check the disposition at the junction of the sarcomeric proteins confirmed the correct expression and targeting to the junction of RyR, DHPR, and triadin (Figure 5B–D, respectively), despite the absence of CASQ1 (Figure 5A). The expression patterns of all proteins were the same as in WT fibers at both sides of the Z-line. Nevertheless, the double-stranded pattern on both sides of the Z-lines, typical of RyR and DHPR staining, was somehow altered; the longitudinal junctions that persisted in adult fibers in CASQ1-null mice, as seen via EM (Figure 5I), were also visible via immunohistochemistry at higher magnification (Figure 5G,H). The longitudinal junctions visible in CASQ1-null fibers were absent in WT fibers (compare Figure 5E,F with Figure 5G,H).

### 3.4. Junctional Protein Expression during Differentiation and Maturation Process in CASQ1-Null Fast-Twitch Fiber

To complement the morphological analyses and further characterize the differentiation process, we analyzed the expression of the key muscle genes expressed during the early and later stages of muscle differentiation. mRNA and protein expression were determined via real-time quantitative reverse transcription-PCR (qRT-PCR, Figure 6) and Western blot (Figure 7), respectively. 

It is well known that T-tubule biogenesis and organization during development occur through differential steps that involve different junctional proteins, such as JP1, that may contribute to the assembly of the multi-protein complex localized at the junction (for a review, see Barone et al., 2015 [13]) CAV3 and BIN1, which contribute to plasma membrane invagination and thus to T-tubule biogenesis [8,9]. Our results showed that in WT muscles, JP1 (the skeletal isoform of junctophilin) reached its higher expression at 1 week of age, while in CASQ1-null fibers, its expression underwent a dramatic increase from 1 week up to 4 months of age (Figure 6A), as also assessed by Western blotting (see below Figure 7A). The increase in JP1 expression was also accompanied by a concomitant increase in expression of BIN1 from 1 week to 4 months of age, as expected, since both proteins play a key role in the correct assembly of CRU during differentiation (Figure 6B).

Validation of BIN1 gene expression came from protein expression analysis using Western blot (Figure 7B). At 4 months of age, CASQ1-null fast-twitch fibers were characterized by a striking proliferation of junctional SR and T-tubule domains, resulting in a dramatic increase in multilayered junctions in place of the classical triads typical of WT muscles. Analysis of gene expression, as well as of protein synthesis in Western blot, of MG29 and CAV3, both playing a dominant role in the T-tubule elongation and formation, showed a significant increase in CASQ1-null fibers compared to WT (Figure 6C,D for qRT-PCR and Figure 7C,D for Western blot analysis).

## 4. Discussion

Postnatal development of CRUs in mouse skeletal muscle fibers undergoes different stages of maturation of the two membrane systems involved in the EC coupling mechanism: the T-tubule and the SR. The process begins during gestation early in myogenesis (E14), and full maturation is achieved around three weeks of age after birth [13,42]. The T-tubule network infiltrates the muscle fiber, and simultaneously, starting about 16 days after birth [41], it changes its orientation from longitudinal to transverse. This process induces CRU localization at the A–I junction, in correspondence with sites where it associates with the SR to form a junctional domain typical of adult fibers (triads), since the SR matures concomitantly with the T-tubule network. Because of the development of the two membrane systems, the appearance of triads is accompanied by a gradual disappearance of peripheral junctions characteristic of the early stages of maturation. The development of specialized junctional domains is coupled with the concomitant assembly of minor and major components of the junctional system: calsequestrin (CASQ), ryanodine receptor (RyR), triadin (TRN), junctophilin 1 (JP1), and caveolin 3 (CAV3) in the SR membrane; and dihydropyridine receptor (DHPR), amphiphysin 2 (BIN1), and mitsugumin 29 (MG29) in the T-tubule membrane. Even though the L-type calcium channel (DHPR) and the calcium release channel (RyR) represent the main actors of the EC coupling mechanism, they require the presence of the other accessory proteins to form the macromolecular complex necessary to finely regulate their correct assembly to junctional domains and their functionality [44]. During development, three proteins are fundamental for T-tubule invagination and elongation inside the muscle fibers: (a) CAV3, which initiates the sarcolemma invagination necessary for T-tubule development; (b) MG29, which forces T-tubule elongation, facilitating the correct conformation for triadic junctions; and (c) JP1, which creates the bridge between the SR and T-tubule membranes and subsequently allows for close and parallel positioning of the two membrane systems in the triad, contributing to the creation of adult CRUs [11,13,44]. Furthermore, JP1 is also found throughout the triad junction when RyR1 is present [45], interacting with it [46]. Over the years it has been demonstrated that creations of knockout mice for one of these three proteins cause morphological alterations in the assembly of CRUs. In particular, lack of CAV3 affects the correct maturation of a highly organized T-tubule system [8], while the absence of MG29 affects the correct alignment of the SR and T-tubule membrane systems necessary for triad formation [10]. Playing a role in the formation of triads in developing skeletal muscle, the lack of JP1 directly impacts the rapid increase in the number of junctions after birth visible in WT muscles [47].

Fast-twitch skeletal muscle fibers from CASQ1-null adult mice show significant morphological alterations in the CRU ultrastructure [35]. The present study aimed to investigate the possible molecular mechanism responsible for triad rearrangement in the absence of CASQ1 in skeletal muscle. Here, we report that the postnatal development of CASQ1-null EDL fast-twitch fibers is characterized by a delay compared to WT in the correct junctional assembly. During development, CASQ expression in CASQ1-null fibers resembles that of WT muscles, with no detectable signs of compensatory increases in CASQ2 expression. Nevertheless, the expression levels of proteins involved in the organization of junctional domains undergo a significant delay compared to WT. The role of JP1 in the formation of triads is not altered by the absence of CASQ1, since they appear correctly positioned within the sarcomere at the A–I junction, even at 1 week of age, and interaction with the other junctional proteins is not altered. On the other hand, only at 1 month of age do CASQ1-null muscle fibers appear to be almost fully developed, with a consistent delay compared to WT. The JP1 expression levels are higher, in correspondence with the developmental stage where full maturation is almost reached, which corresponds to 1 week for WT fibers and 1 month for CASQ1-null fibers, when the increase in the number of multiple junctions is visible. JP1-overexpression in CASQ1-null fibers suggests a role in the maturation and maintenance of triads, considering that JP1 knockout is responsible for the reduction in the number of triads and abnormalities in the SR [14]. It has been demonstrated that the BIN1 N-terminal domain induces membrane curvature during junction assembly [9,48]. Hence, the increased expression of BIN1at the same developmental stage in which T-tubule starts remodeling validates its crucial involvement in the recruitment of membranes for EC coupling-apparatus formation. Formation of deep membrane infoldings within the muscle membrane due to BIN1 expression that leads to maturation of adult CRUs and structural integrity of the multilayered structures typical of adult CASQ1-null skeletal muscle fibers guaranteed by JP1 is supported by MG29 and CAV3 overexpression, which in WT muscle under physiological conditions is mainly expressed early during myogenesis, even before triad formation. The expression of MG29 and CAV3 in adult fibers ensures proper Ca^2+^ handling and muscle contraction because they are, respectively, involved in the organization and maintenance of CRUs and in the formation of caveolae for the assembly of signaling molecules.

Over the years, several knock-down or knock-out mouse models of junctional proteins have demonstrated their crucial roles in the correct assembly and maturation of junctional membrane domains [8,15,49,50]. Here, we show that the ablation of CASQ1 drives a delay in the maturation process and a morphological rearrangement of junctions as a possible consequence of the overexpression of proteins in the late stages of development. As a compensatory mechanism, significant ultrastructure changes, i.e., multilayered junctions, are caused by the lack of CASQ1. Furthermore, an increase in CRU formations is accompanied by the consequent increase in the number of SR Ca^2+^-release channels and RyR1s [35]. 

## 5. Conclusions

The present study comes as a natural consequence of the first characterization of CASQ1-null skeletal muscle fibers (Paolini et al., 2007) [36] in order to identify the possible mechanism responsible for impairment of the expression and accumulation of junctional proteins, resulting in an adaptation of the EC coupling apparatus necessary to maintain physiological function. The consequent ultrastructural remodeling resembles the condition of a delayed and incomplete development process of CRUs.

## Figures and Tables

**Figure 1 biology-12-01064-f001:**
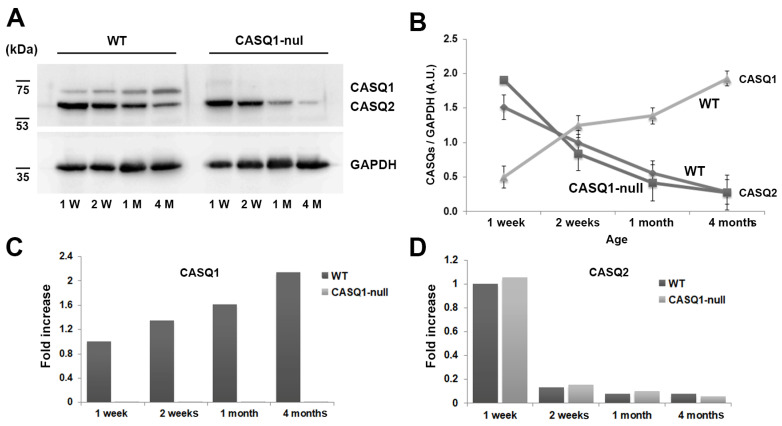
Expression levels of CASQ1 and CASQ2 in WT and CASQ1-null EDL muscle. (**A**) Representative immunoblot of the expression levels of CASQ1, CASQ2, and GAPDH at different ages in EDL muscle. (**B**) Densitometric analysis of immunoblots is given as relative amount of CASQ normalized to GAPDH in WT and CASQ1-null EDL muscles (n = 3 muscles from 3 different mice for each group). (**C**,**D**) Relative quantification of CASQ1 and CASQ2 mRNA levels (pool of 10 EDL muscles/group) as determined by real-time PCR, following normalization to GAPDH mRNA.

**Figure 2 biology-12-01064-f002:**
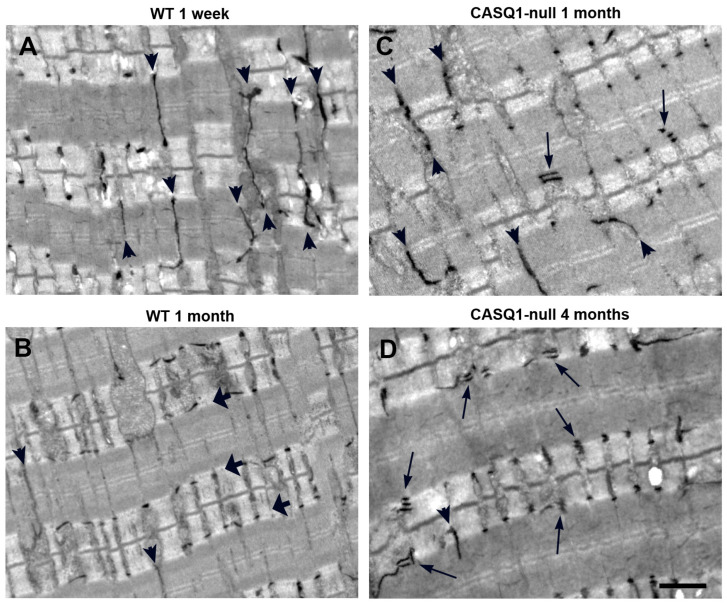
Staining of T-tubular network in WT and CASQ1-null EDL fibers. (**A**,**B**) T-tubule network stained with ferrocyanide at 1 week and 1 month of age in WT EDL fibers. Small arrows in panels (**A**,**B**) point to longitudinally oriented tubules; larger arrows in panel (**B**) point to T-tubules that reached their final transversal positioning at the A–I band transition. (**C**,**D**) T-tubule network stained with ferrocyanide at 1 and 4 months of age in CASQ1-null EDL fibers. Small, thick arrows point to longitudinally oriented tubules; long, thin arrows point to multiple T-tubules in close proximity, indicating the presence of CRUs formed by multiple elements. Bars: A–D, 1 µm.

**Figure 3 biology-12-01064-f003:**
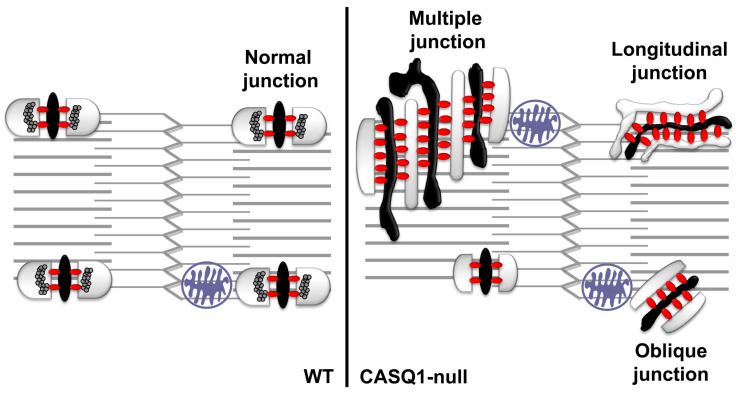
Model of different types of CRUs in adult WT fibers (**left**) and CASQ1-null fibers (**right**).

**Figure 4 biology-12-01064-f004:**
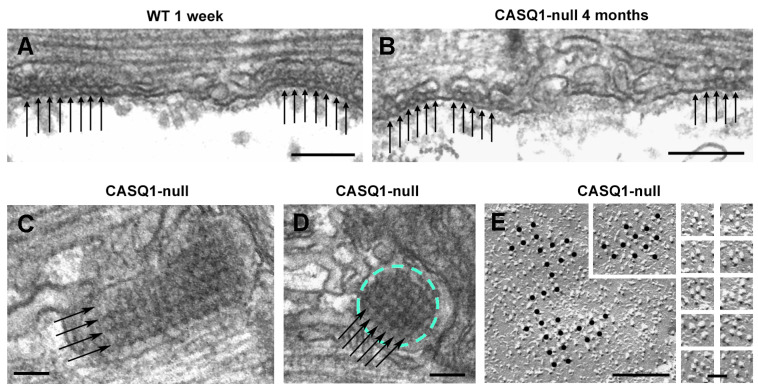
Peripheral couplings are frequent in adult CASQ1-null fibers. (**A**,**B**) Peripheral couplings in WT muscle (at 1 week of age) and in CASQ1-null fibers (at 4 months of age). Arrows in panels (**A**,**B**) point to electron densities representing RyRs/feet. (**C**,**D**) Arrays of RyRs/feet forming multiple rows (arrows) in internal junctions. (**E**) Freeze fractures of the surface membrane showing DHPRs in peripheral couplings organized in tetrad arrays (see the insets for details). Bars: (**A**,**B**), 0.2 µm; (**C**,**D**), 0.1 µm; (**E**), 0.25 μm; gallery of images, 0.1 μm.

**Figure 5 biology-12-01064-f005:**
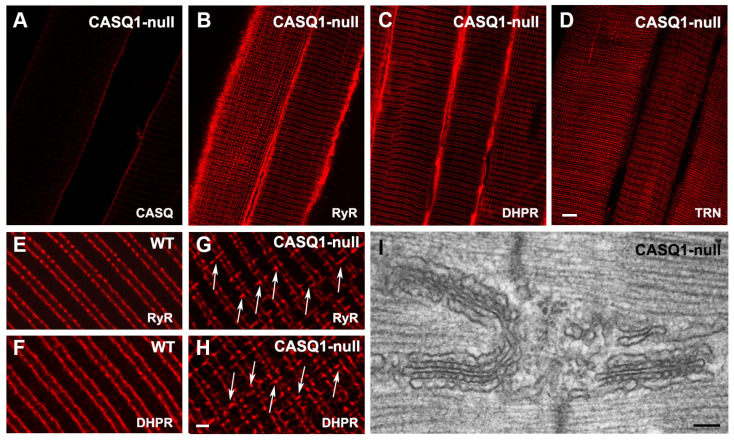
Immunolabeling of triadic proteins (and EM of longitudinal triads) in WT and CASQ1-null adult (4-month-old) fibers. (**A**–**D**) Confocal images of CASQ1-null EDL fibers immunolabeled with antibodies against RyR, DHPR, and triadin (TRN). (**E**–**H**) Higher magnification confocal images: arrows in (**G**,**H**) point to immuno-positive RyR and DHPR foci longitudinally oriented, which are not present on WT. (**I**) EM of longitudinally oriented triads in a CASQ1-null fiber. Bars: (**A**–**D**), 5 µm; (**E**–**F**), 1 mm; (**I**), 0.1 µm.

**Figure 6 biology-12-01064-f006:**
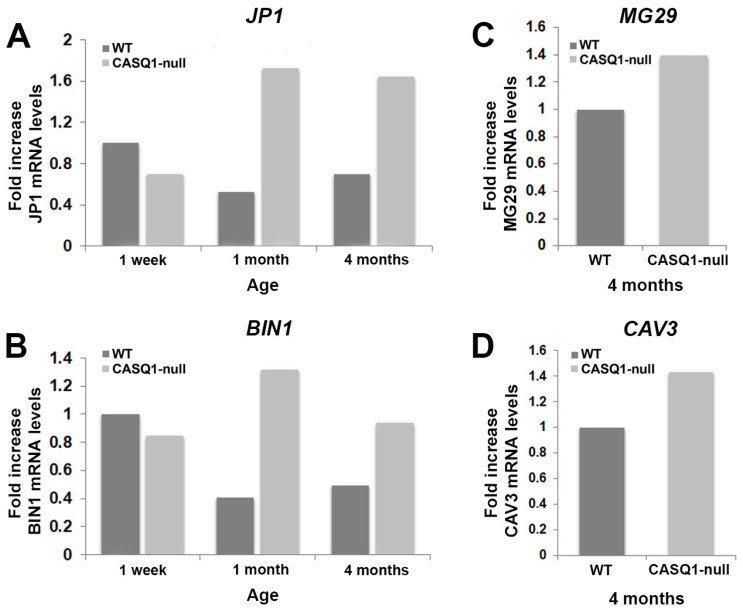
RT-PCR analysis of mRNA transcripts (JP1, BIN1, MG29, and CAV3) in WT and CASQ1-null mice during development. (**A**,**B**) Analysis of mRNA transcripts of junctophilin 1 (JP1) and amphiphysin 2/Bridging Integrator-1 (BIN1) from CASQ1-null EDL muscles at different postnatal stages (light grey bars), expressed as fold change relative to WT EDL muscles at 1 week of age (dark grey bars). (**C**,**D**) Analysis of mRNA transcripts of mitsugumin 29 (MG29) and caveolin 3 (CAV3) in CASQ1-null adult EDL muscles (light grey bars), expressed as fold change relative to WT EDL muscles (dark grey bars).

**Figure 7 biology-12-01064-f007:**
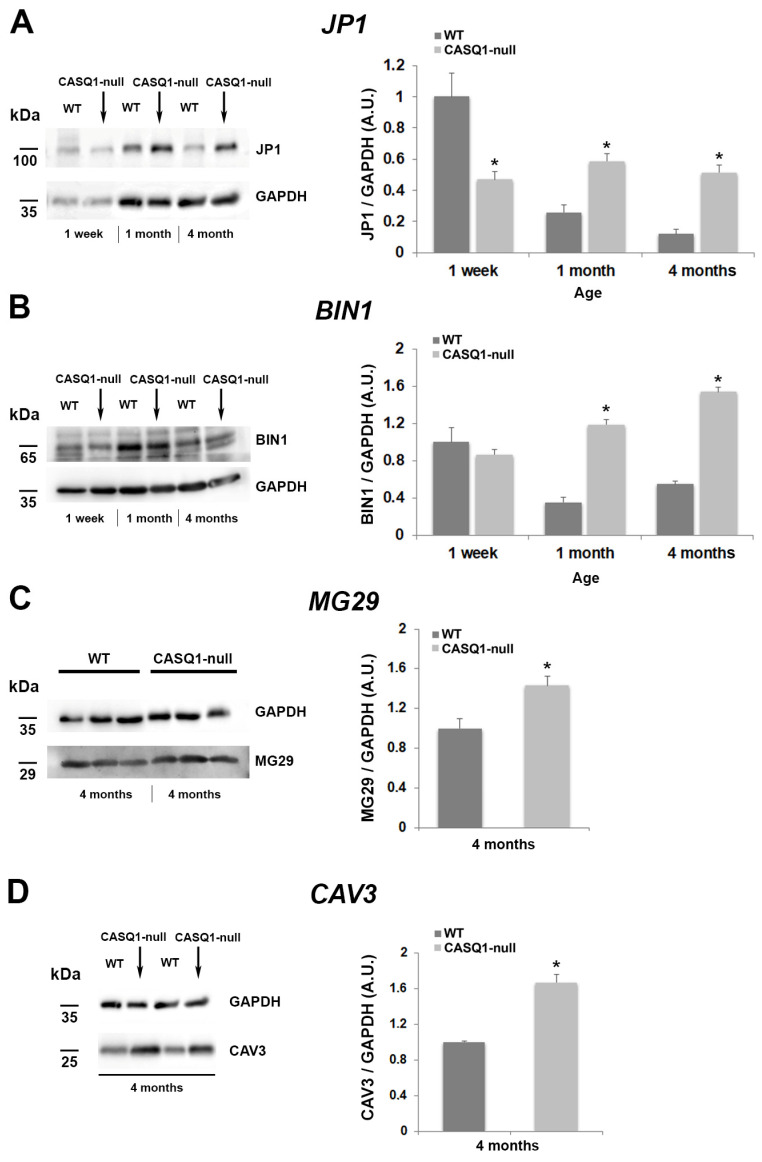
Western blot analyses of JP1, BIN1, MG29, and CAV3 in WT and CASQ1-null mice during development. (**A**–**D**) Representative immunoblots showing expression levels of JP1 and BIN1 in EDL muscle homogenates from WT and CASQ1-null mice during development (from 1 week to 4 months of age). (**C**,**D**) Expression levels of MG29 and CAV3 in EDL muscle homogenates from WT and CASQ1-null mice at 4 months of age. Data are expressed relative to WT EDL muscles at 1 week of age (**A**,**B**) and to WT EDL muscles at 4 months of age (**C**,**D**). Data are shown as mean ± SEM; * *p* < 0.05. n = 6 muscles from 6 different mice for each group.

**Table 1 biology-12-01064-t001:** Quantitative analysis of the frequency during development (1 and 4 months of age) of the different type of CRUs in WT and CASQ1-null fibers. Data are means ± S.D. n = number of fibers analyzed. * Significantly different from WT group at *p* < 0.01.

EDL	No. Junction/100 μm^2^	No. Multiple Junctions/100 μm^2^	No. Longitudinal Junctions/100 μm^2^	No. Oblique Junctions/100 μm^2^	No. Transversal Triads/100 μm^2^
WT 1 month	106.2 (n = 118)	1.7 ± 3.3 (1.6%)	9.5 ± 11.9 (8.9%)	11.50 ± 8.6 (10.8%)	83.5 ± 27.1 (78.6%)
CASQ1-null 1 month	89.4 (n = 123)	5.7 ± 6.5 * (6.3%)	19.1 ± 10.1 * (21.4%)	11.2 ± 7.1 (12.5%)	53.4 ± 21.3 * (59.7%)
CASQ1-null 4 month	106.0 (n = 146)	23.66 ± 4.8 * (22.0%)	24.06 ± 11.1 * (22.7%)	6.01 ± 2.0 (5.7%)	52.3 ± 12.8 * (49.3%)

## Data Availability

Not applicable.

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
