# Peer review of "Structural Adaptation of the Excitation–Contraction Coupling Apparatus in Calsequestrin1-Null Mice during Postnatal Development"

_biology, 2023, doi:10.3390/biology12081064_

Round 1

Reviewer 1 Report

The manuscript presents a detailed analysis of the ultrastructural alteration of the sarcotubular system in the EDL muscle of CASQ1-null mice. The analysis was performed during the course of muscle development, at 1 week, 1 month and 4 months of age both on the structure and the protein expression level. The authors concluded that the absence of CASQ1 leads to several defects of the t-tubule orientation (longitudinal and oblique) as well the remaining presence of peripheral coupling at 4 months. They reported also a delay expression of several junctional proteins implicated in the formation/maturation of the sarcotubular system such as JP1 or BIN1. This paper extends previous findings from the same team published in 2007 (Paolini et al. J. Physiol) with more time points analyzed and the expression of junctional proteins. The EM pictures and their analysis are convincing but the new pieces of information are scarce.   

Specifically:

The way the results regarding protein expression are presented and discussed is confusing. For instance, lines 429-430: it is written that the expression of proteins involved in junctional assembly is delayed. This is not clear from figure 6, where indeed there is a reduced expression level at 1 week but this is not anymore the case at 1 and 4 months. In addition, the authors talk about JP1 expression levels that drastically increased at 1 month. To me, increase refers to a comparison with 1 week, while for the authors it is an increase compared to WT. Comparing to 1 week, the level of JP1 did not increase. Please clarify.

Lines 363-365 : “The increase in JP1 expression is also accompanied by a concomitant increased expression of BIN1 from 1 week to 4 months of age, confirming the different role of both protein in the correct CRU assembling during differentiation “ I don’t see the logic of this sentence. Why having concomitant increased of expression confirm that both proteins have different roles?

Figure 5: The results are presented normalized to WT at one week, correct? It has to be mentioned (one week is missing). There is no statistical analysis for all data of figure 5. It should be included.

Figure 6: it is not stated that the data are normalized, obviously to 1 week WT. Should be added.

In the discussion, there is a lack of perspectives to connect the morphological alterations observed, with functional consequences for muscle contraction and/or EC coupling. As well, the novelty compared to the previous paper of the same group (Paolini 2007) should be better highlighted.  

Minor :

Fig 2 and 3. The labelling in black is poorly visible, try in white to make it clearer.

Line 231 : “WT e CASQ1-null fast-twitch fibers” should be: “WT and CASQ1”

The cartoon of page 7 is labelled as table 1. This is strange. As well, this cartoon might well be the last figure of the paper, as it sum up the main findings, no?

Table 1: there is no * on the results, but only one data highlighted in bold? Significant results must be indicated.

There is no value for the condition CASQ1-null 4 month and the n° of oblique junctions. Why? Is the value 0, or was is not analyzed? On line 289 it is stated that the number of oblique junctions decreases? Please comment.

Figure 3: the scale bar is missing for panel E

Nothing to say

Reviewer 2 Report

Murzilli and coworkers have conducted an interesting study showing the functional importance of CASQ1 for normal development of E-C coupling apparatus. The manuscript is well written with clear presentation of the data. A combination of morphological and biochemical analyses is a particular strength of the study.

Comments:

·        Figure 1A: Figure legends say that CASQ1 and 2 were normalized to GAPDH. However, this is probably not what is meant in this case since only a picture of the blot is presented. Perhaps data were normalized in Figure 1B?

·        Figure 1B-D: Y-axes contain numbers with decimal commas, which should be corrected to decimal dots.

·        Figure 1D: How many measurements were done to construct graphs in Figure 1C and 1D? For the reasons of clarity and transparency, it would be better if dots are used to indicate individual measurements in Figure 1C-D.

·        The cartoon on page 7 appears to have a wrong title (Table 1). To make the scene even more useful to the readers A- and I-bands should be indicated. The cartoon/scheme should be referred to in the text.

·        Table 1 is very interesting. Perhaps the authors would consider reporting also the fraction or percentage of multiple, longitudinal, and oblique junctions etc.? Since the total number of junctions is not the same between WT and KO mice (at 1 month and 4 months), reporting the fractional frequency would useful to fully appreciate the data.

·        Figure 5: How many measurements (from how many muscles/mice) were performed to construct graphs 5A-D? Effects seem robust, but were differences statistically significant?

·        Figures 5 and 6: decimal commas should be replaced by decimal dots.

·        Figure 6:  How many measurements (from how many muscles/mice) were performed to construct graphs 6A-D?

·        Line 452: “as a consequence of an overexpression of proteins…”. It would probably be better to say as a possible consequence. Despite the apparent correlation between the protein abundance and developmental delay, the causal link has not been demonstrated experimentally in this study.

·        Line 426: “The present this study…”: “this” is not needed here.

·        Perhaps semi-quantitative PCR (line 241) is not the best choice of words for the method used here.

Reviewer 3 Report

Manuscript [biology-2508597] entitled ‘Structural adaptation of the excitation-contraction coupling apparatus in Calsequestrin1-null mice during postnatal development’ by Murzilli and colleagues focuses on the characterization of CASQ1 KO mice and the effects on the EC-coupling and calcium release units during development and maturation. This is an interesting study on the central role of CSQ in EC-coupling and CRUs. The findings from the ultrastructural and immunolabelling analyses of CSQ1 KO muscles are well presented.

The authors are encouraged to address the following issues:

(i) Abstract section: The abstract uses non-standard abbreviations, such as ‘CRUs’. Abbreviations should best be introduced at first usage, as in the Introduction section. An abstract should be standalone and ideally not use non-standard abbreviations. Please see the author’s instruction for the policy of this particular journal on the usage of abbreviations in the abstract section. 

(ii) Abstract: Why is the term ‘WT’ for wild type mentioned, but then not used a second time in the abstract. 

(iii) Abstract: The same is true for the abbreviations of proteins such as RYR, DHPR, JP1, BIN1, CAV3 and MG29 in the abstract section. Ideally, an abstract should avoid too many abbreviations, but if necessary, then they have to be explained. 

(iv) Introduction (line 46): The introductory sentence ‘Excitation-contraction (EC) coupling, the process that controls release of Ca2+ during muscle activation [1,2], occurs at …’ is confusing. Physiologically, EC-coupling does not control calcium release, but the temporal and spatial changes in calcium cycling regulates EC-coupling. Please re-word this sentence to properly reflect the physiological mechanisms that underlie the regulation of EC-coupling and calcium handling in skeletal muscles. In addition, the term ‘muscle activation’ is not clear. ‘Activation’ can mean various processes in biology. Please state clearly that ‘muscle contraction’ is initiated by this regulatory process.

(v) Introduction (first paragraph): please clearly state that the description of the developmental steps relate to ‘mouse’ muscle.

(vi) Introduction (third paragraph): The description of the differential and developmental expression of ‘fast’ CSQ1 versus ‘slow/cardiac’ CSQ2 is mostly based on biochemical and cell biological findings in this manuscript. However, more sensitive biochemical methods usually identify both CSQ isoforms in mature muscles by mass spectrometry. MS-based proteomic studies, which have for example been used to demonstrate changes in the expression of CSQ1 (10.1152/japplphysiol.00911.2013), routinely detect both CSQ isoforms in skeletal muscle. This should ideally be mentioned in the overview on CSQ isoform expression patterns. See for example: Deshmukh et al. Deep proteomics of mouse skeletal muscle enables quantitation of protein isoforms, metabolic pathways, and transcription factors. Mol Cell Proteomics. 2015;14(4):841-53 (10.1074/mcp.M114.044222). In the consensus MS data files (presented as supplementary tables) in both muscle tissue and muscle cell culture preparations, both isoforms, CSQ1 and CSQ2, are routinely identified by sensitive MS analysis. This should be mentioned in this paper and clearly state that highly sensitive proteomics detects both CSQ isoforms in mature skeletal muscles.

(vii) Introduction (line 99): Can the authors please elaborate already in the Introduction section briefly on this very interesting observation that ‘… lack of CASQ1 causes striking remodeling of CRUs (more evident in fast twitch than in slow twitch muscles) with a consequent partial impairment of the EC coupling mechanism …’. Why is there such a striking difference between fast and slow muscles at the level of CRUs and is this directly related to the differential expression of CSQ1 versus CSQ2 isoforms in wt muscles?

(viii) Immunoblotting with antibodies to calsequestrin isoforms (figures and original images): Can the authors please outline whether mouse CSQs are also recognized as several high-molecular-mass bands, the so-called CSQ-like proteins, besides the main 45 kDa band, by their antibodies? In both human and rabbit muscle, anti-CSQ antibodies usually recognize the main CSQ monomer and several additional bands of higher molecular mass. Is this different in mouse muscle?

(ix) Can the authors please outline in more critical detail the underlying objective why the EDL was chosen for their analyses and the focus on fast fibres. Is the fibre type distribution affected by the CSQ1-KO process? How are other types of skeletal muscles affected by the loss of CSQ1 and what compensatory mechanisms, other than RyR1s, are known that may act in a fibre type-specific manner?

(x) SERCA calcium pumps: CSQ functioning is closely linked to the RyR-CRC units, as outlined in this report. However, it is surprising that the authors did not discuss in more critical detail the potential effects of CSQ1 KO on the calcium re-uptake apparatus, which is crucial during muscle relaxation and thus the overall regulation of calcium homeostasis and muscle function. EC-coupling/calcium-release and subsequent calcium re-uptake during relaxation should really be seen as very closely related physiological processes and this discussed in this paper n the effects of CSQ1 KO. Is anything known about potential changes in the expression profile of the calcium pumps (SERCA1, SERCA2 and their many regulators such as phospholamban or sarcolipin) in CSQ1 KO mouse muscle?

(xi) Sarcalumenin, parvalbumin, calmodulin, regucalcin: It is surprising that the authors did not analyse or discuss in more critical detail the potential effects of CSQ1 KO on the luminal SR calcium shuttle and binding protein sarcalumenin, as well as the crucial cytosolic calcium binding proteins calmodulin, parvalbumin and regulcalcin. These key calcium binding proteins of the SR and the cytosol could play major roles in compensatory mechanisms in CSQ1-lacking fibres. The authors are encouraged to discuss the key elements of the entire core muscle calcium handling apparatus in more critical detail.

The general text contains numerous confusing or slightly awkward or grammatically not quite correct sentences. For example, line 14 (… influences the efficiency of fibers contractile machinery …), line 20 (… how an altered progression in their expression during …), line 35 (… observed in CASQ1-null mice hindered the …), line 65 (… finally Junctophilins (JP), which acts as a …), etc. Please check your manuscript for proper wording and sentence structures prior to resubmission. 
